# Polycyclic Aromatic Hydrocarbon Sorption by Functionalized Humic Acids Immobilized in Micro- and Nano-Zeolites

Gabriela Robles-Mora [1], Josefina Barrera-Cortés [1,2,*], Lucila Valdez-Castro [3], Omar Solorza-Feria [1,4] and César García-Díaz [5]

1    Programa de Nanociencias y Nanotecnología, Centro de Investigación y de Estudios Avanzados del Instituto Politécnico Nacional (Cinvestav-IPN), Unidad Zacatenco, Ciudad de México 07360, Mexico; etherob@hotmail.com (G.R.-M.); osolorza@cinvestav.mx (O.S.-F.)
2    Departamento de Biotecnología y Bioingeniería, Cinvestav-IPN, Unidad Zacatenco, Ciudad de México 07360, Mexico
3    Programa Académico de Ingeniería en Biotecnología, Universidad Politécnica de Puebla, Puebla 72640, Mexico; lucila.valdez@uppuebla.edu.mx
4    Departamento de Química, Cinvestav-IPN, Unidad Zacatenco, Ciudad de México 07360, Mexico
5    Tecnológico de Monterrey. Escuela de Ingeniería y Ciencias. Campus Estado de México. Carretera Lago de Guadalupe km 3.5, Atizapán de Zaragoza. C.P. 52926. Estado de México, Mexico; cgarciad@tec.mx
*    Correspondence: jbarrera@cinvestav.mx; Tel.: +52-55-57473800 (ext. 4380)

**Abstract:** Polycyclic aromatic hydrocarbons (PAHs) are hazardous compounds originating from anthropogenic activity. Due to their carcinogenic properties for humans, several technologies have been developed for PAH removal. Sorption with natural and organic materials is currently one of the most studied due to its low cost and its environmentally friendly nature. In this work, a hybrid sorbent involving functionalized humic acids (HAs) and nano-zeolite is proposed to entrap PAHs. The use of functionalized HAs immobilized in a porous support is designed to address the instability of HAs in solution, which has been already reported. HA functionalization was carried out to increase the non-polarity of HAs and aliphatic group formation. The HAs were functionalized by esterification/etherification with alkyl halides, and their chemical changes were verified by FTIR and NMR. The sorption isotherms of the functionalized HAs in micro- and nano-zeolites were used to assess the performance of the nano-zeolites in adsorbing these HAs. The hybrid support allowed the removal of anthracene and pyrene at percentages higher than 90%; fluoranthene, of angular molecular structure, was adsorbed at 85%. PAHs are ubiquitous in the environment, and a stable sorption of them in solid matrices will allow their removal from the environment through effective and environmentally friendly methods.

**Keywords:** chemically modified humic acids; nano-zeolite; micro-zeolite; PAH removal; sorption isotherms

## 1. Introduction

Polycyclic aromatic hydrocarbons (PAHs) are hazardous organic compounds that are ubiquitous in the environment [1]. These compounds originate from anthropogenic activity and natural processes involving the incomplete combustion of organic matter [2]. The hydrophobicity of PAHs favors their adsorption in the organic matter of soils and consequently their transport to different ecosystems [3,4]. Out of 400 reported PAHs, 16 have been classified and identified by the EPA as priority pollutants due to their elevated toxic and carcinogenic properties for humans [2]. For PAH removal from the environment, several remediation technologies have been developed including physical, chemical, and biological methods [3,5].

Adsorption is a technology that allows pollutant removal by physical methods. It is one of the most studied in recent years due to its simplicity and because it does not generate toxic pollutants [6]. A wide variety of both natural and synthetic materials have

been used as sorbents; however, due to their large sorption surface, activated carbons are still the preferred sorbents despite their high costs [4,7]. Cost-efficient materials, such as zeolites and even organic matter itself, are still being researched [8].

Zeolites are microporous, aluminosilicate minerals of negative ionic charge, which, among a wide variety of applications, are used as sorbents to remove toxic pollutants from the environment [9]. Zeolites are characterized by their ion exchange capacity, with high affinity to the cationic form. However, the treatment of zeolites with organic salts or organic surfactants has allowed their interaction with anions and organic compounds such as PAHs [8]. Of the different types of interactions occurring between zeolites and surfactants, ion exchange and hydrophobic bonding are the predominant ones [10]. It has been reported that decreasing the particle diameter represents a strategy to increase the sorption capacity of zeolites [11]. Zeolites are natural minerals from volcanic and rocky areas. However, numerous synthetic zeolites are currently produced for specific applications, mainly for use as catalysts [12].

Seventy percent of the organic matter contained in soils comprises humic substances, and of these, humic acids (HAs) are the predominant fraction at concentrations of around 50% [13]. Agricultural soil fertilization is the main application of HAs. Furthermore, given the heterogeneous molecular structure of these substances, which integrate different functional groups, extensive research is currently being undertaken in order to take advantage of their noteworthy properties, which include the sorption of organic pollutants [4,14,15].

HAs are known to adsorb PAHs contained in both water and soils, and the PAH structure and the HA number of active functional groups are determining factors in the magnitude of the interaction between these two components [4]. The number of active sites has been successfully modified through esterification–etherification reactions. The use of these functionalized HAs in washing soil impacted with crude oil increased the hydrocarbon removal up to 28% [16]. PAH sorption in HAs has been attributed to π–π interactions, hydrophobic effects, and hydrogen bonds, independently of whether the adsorption process is carried out in the presence of HA in solution, in its solid form, or immobilized in a microporous solid [16,17]. It is assumed that an immobilized HA should generate a more effective pollutant sorption considering the instability of HAs in solution [15].

The objective of this work was to evaluate the capacity of functionalized HAs immobilized in zeolite to adsorb selected PAHs. The sorption capacity depends on the active surface available [11]. Therefore, the functionalized HAs were immobilized on micro- and nano-zeolites. HA functionalization was carried out chemically by esterification and etherification using alkyl halides. Leonardite, produced in the State of San Luis Potosí, Mexico, and HAs used as fertilizers for agricultural soils were the source of humic acids. Using HAs contained in fertilizers allows us to gain knowledge of the activity that the materials commonly dispersed in the environment could have in relation to those expressly designed for application in the remediation of polluted soils. Furthermore, immobilizing humic acids on insoluble supports gives the possibility of physically separating the supports and, consequently, the contaminants trapped in them.

## 2. Materials and Methods

### 2.1. Source and Purification of Humic Acids

Humic products offered commercially for soil fertilization were used in this research: Humin Tech Pow Humus WSG with 85% humic acids (HTPH) and leonardite (LEO-MX) donated by the Mexican company Tecnología Especializada en el Medio Ambiente Available online: (https://temamx.com.mx/). The humic acids extracted from these materials were identified as HA-TPH and HA-MX, respectively. The HAs were extracted in batches of 2 g following a conventional procedure recommended by the International Humic Substances Society [18].

Humic compound samples (2 g) were poured into 10 mL of a (1:1) solution prepared with NaOH (0.1 N) and $Na_4P_2O_7$ (0.1 M) to solubilize HAs at 200 rpm for 12 h under a

nitrogen atmosphere. Non-soluble matter was separated by centrifugation (10,000 rpm for 15 min) and the supernatant acidified (pH = 1) with HCL (6 M) to precipitate HAs for 12 h under stirring (200 rpm). HAs were separated by centrifugation (10,000 rpm for 15 min), and next, 10 mL of a (1:1) HCl (0.5%)/HF solution (2 g per each 10 mL) were added to eliminate possible silicates contained in the HAs, for 12 h at room temperature and with stirring (200 rpm). HAs were separated by centrifugation and the chlorine ions removed by dialysis (molecular porous membrane tubing of 6–8 kDa, 1 Spectra/Por Dialysis Membrane, Sprectrum Labs Inc. USA) with deionized water until the electrical conductivity of this was kept under 10 μS for more than 24 h. The HAs were dried at 50 °C and stored in vials at room temperature.

### 2.2. Functionalization of Humic Acids

Humic acid functionalization was carried out in triplicate by etherification and esterification from O-alkylation reactions, using tetrabutylammonium hydroxide (TBAH: $(C_4H_9)_4NOH$) as a catalyst [16]. Purified HA (500 mg) and a prepared solution containing 1.1 mL of 1.2 M NaOH, 10 mL of distilled water, 200 μL of TBAH, and 300 μL of an alkyl halide (PBr, pentyl bromide ($CH_3(CH_2)_4Br$); IM, iodomethane ($CH_3I$); or BBr, benzyl bromide ($C_7H_7Br$)) were combined in a 250 mL Erlenmeyer flask and mixed at 200 rpm and 45 °C for 24 h. The precipitated solids were separated by centrifugation (10,000 rpm, 5 min) and washed repeatedly with acidified distilled water (pH = 1). The separated HAs were dried at 50 °C for 24–48 h [16]. All chemical compounds were purchased from Sigma-Aldrich Mexico.

The HA chemical modification was verified by FTIR and NMR spectra. FTIR spectra were assessed for a KBr pellet by means of a Nicolet 6700 FTIR spectrophotometer (Thermo Fisher Scientific USA) implemented with the Omnic 6.0 software. Pellets were obtained by accurately weighing 3 mg HA and 100 mg of KBr, both dried and ground in an agate mortar. The recorded spectra were in the 4000 to 400 cm$^{-1}$ range with a resolution of 2 cm$^{-1}$, and each sample was scanned 64 times. The NMR data were acquired using a ramp cross-polarization pulse program with magic angle spinning on a Bruker Avance II 300 MHz NMR spectrometer. Spectra were acquired at a frequency of 75 MHz for $^{13}C$ and 300 MHz for $^1H$ with a CPMAS spinning rate of 10 kHz, 2 ms contact time, 5 s recycle delay, more than 10,000 scans per sample, and line broadening of exponential multiplication with a factor of 20. Samples were tightly packed into 4 mm zirconia rotors.

### 2.3. Characterization of Humic Acids

Characterization of the purified and functionalized HAs consisted of moisture content, ash content, HA maturity, solubility, superficial tension, and elemental analysis. The elemental analysis (total carbon (C), hydrogen ($H_2$), and nitrogen ($N_2$)) was performed on a Thermo Finnigan FlashEA® 1112. Humic acid moisture was determined by heating HA samples at 105 °C for 24 h. The ash content was determined by burning samples (0.5 mg) at 700 °C for 4 h. The HA maturity was determined by the UV–Vis spectra of HA solutions (2 mg/25 mL 0.05 N NaHCO$_3$) at 465 and 665 nm wavelengths (E4/E6) (Thermo Spectronic Genesys 10 UV) [19]. The HA solubility was determined by pouring HA samples (0.1 g) into 30 mL of distilled water contained in 100 mL Erlenmeyer flasks, and then adding a 1 M NaOH solution dropwise until all HA solids had dissolved. The surface tension of the corresponding HA solutions was measured with a TD1C LAUDA tensiometer.

### 2.4. Immobilization in Zeolite of Humic Acids

The HAs were immobilized in <45 μm zeolite acquired from Sigma-Aldrich Mexico (No. 96096) and ground zeolite of 170 nm mean diameter. The zeolite milling was performed in a PM 400 planetary ball mill (zirconia ball diameter of 2 cm) at 200 rpm for 96 h. The particle diameter was determined using a high-resolution scanning electron microscope (FE HRSEM Auriga 3916 Zeiss Microscope, Oberkochen, Baden-Wurtemberg, Germany) with a Schottky-type field emission electron source, GEMINI column 1 nm

at 15 kV and 1.9 nm at 1 kV. The mean particle size was determined by image analysis (SmartSem software V05.06) and the zeolite elemental composition by the point mapping of elements (Bruker Quantax 200 Energy Dispersive X- ray Spectrometer EDS). The specific surface of the ground and unground zeolite was determined by the BET method.

Zeolite samples of 0.2 g were poured into Erlenmeyer flasks of 125 mL containing 40 mL of an HA solution of 10, 25, 50, or 150 mg L$^{-1}$ and with a pH of 0.5 (or 3.5) [20–22]. All samples were left to adsorb for 72 h at 25 °C and 200 rpm for the two zeolite particle sizes. The HA sorbed into the zeolite ($Q_e$) was calculated by means of the following equation [23,24]:

$$Q_e = \frac{(C_0 - Ce)V}{m} \tag{1}$$

where $V$ is the volume (L) of the HA solution, $C_0$ is the initial concentration of the HA solutions (mg L$^{-1}$), $C_e$ (mg L$^{-1}$) is the HA concentration at the solid–liquid equilibrium, and $m$ is the weight (g) of the water-free zeolite. The $C_e$ was determined from absorbance measurements at 285 nm (Thermo Spectronic Genesys 10 UV) of the HA solid particle free solution. The HA concentration (mg L$^{-1}$) was determined by interpolation in a calibration curve of HA concentration vs. absorbance.

### 2.5. Sorption of PAHs into Pure and Chemically Modified HA

Naphthalene (NAPH), fluoranthene (FLU), anthracene (ANT), and pyrene (PYR), acquired from Sigma-Aldrich Mexico, were the polycyclic aromatic hydrocarbons (PAHs) used in this test. Sorption of these PAHs in the pure and modified HA was left in liquid phase for 72 h with stirring (200 rpm) at a controlled temperature of 25 °C. PAHs were added to the HA solutions at a concentration of 30 ppm. The concentration and pH of the HA solutions were dependent on the results obtained in the previous section. The solids were separated by centrifugation at 10,000 rpm for 5 min and subsequently washed with acidified distilled water (pH = 1). The solids were dried in heat atmosphere at 40 °C for further analysis of functional groups by FTIR.

### 2.6. Sorption of PAHs in an HA-Zeolite Hybrid Sorbent

The PAHs were sorbed in an HA-zeolite (HA-ZEO) hybrid sorbent selected according to the results obtained in Section 2.4. Half a milliliter of a known PAH solution dissolved in dichloromethane was poured into a 125 mL Erlenmeyer flask. After solvent evaporation at room temperature, the HA-ZEO hybrid sorbent was added suspended in 40 mL for PAH sorption at 25 °C and 200 rpm for 72 h. PAH concentration in this suspension was 30 mg L$^{-1}$. Non-sorbed PAHs were recovered by liquid–liquid extraction with dichloromethane following separation by centrifugation (10,000 rpm for 5 min) of the solid particles. The PAH concentration was determined by gas chromatography (GC) in a PerkinElmer Clarus 500 chromatograph with a Thermo Fisher Scientific 100% dimethyl polysiloxane (30 mL × 0.25 mm ID ×1.0 μm) column. The detector and injector temperatures were set at 290 and 320 °C, respectively, and the carrier gas (N$_2$) at 50 mL/min. The oven was operated in a 65 to 310 °C range, according to the following temperature program: 65 °C for 3 min; 25 °C/min until 180 °C; 10 °C/min until 280 °C; 5 °C/min to 310 °C [25].

The percentage of sorbed PAHs in the HA-ZEO hybrid sorbent was determined from the following equation [23]:

$$\%PAH_{entrapped} = \frac{(A_{control} - A_{supernatant})}{A_{control}} \times 100 \tag{2}$$

where Acontrol is the area under the peak of the PAH extracted from the control trial implemented at 30 ppm concentration without the hybrid sorbent, and Asupernatant is the peak area of PAH extracted from the supernatant.

*2.7. Statistical Analysis*

All the experiments were performed in duplicate, as well as the experimental design. The different capacity to sorb PAHs of the tested humic acids and the HA-ZEO solid matrix was determined by an ANOVA for a $p(F) < 0.05$ (Minitab 18).

## 3. Results

*3.1. Elemental Composition of Humic Substances*

The humic acids extracted from Mexican leonardite (LEO-MX) and the commercial Humin Tech Pow Humus (HTPH) product were recovered at percentages of $14 \pm 3\%$ (HA-MX) and $33 \pm 2\%$ (HA-TPH), respectively. The elemental composition (C, H, and N) in these humic substances and their respective pure and functionalized HAs is shown in Table 1. These elements were found in higher concentrations (65–70%) in the HTPH product. However, after the HA extraction and purification, the C and N content was higher by 28% and 9%, respectively, in the HA-MX. Functionalized HAs exhibited an additional increment of the three elements at a percentage that was dependent on the source of the pure HA and the reactive agent. In the case of the HA-MX functionalized with MI, the C, H, and N percentages increased by 8%, 44%, and 28%, respectively. In the HA-TPH, C content increased by 42.2% with the BBr and the H and N elements by 92.7% and 62.3%, respectively, with the PBr. According to the observed increments, the HA-TPH was more susceptible to chemical modification with the longest carbon chain alkyl halides.

**Table 1.** Carbon (C), hydrogen (H), and nitrogen (N) content in the two HA sources (LEO-MX and HTPH), and in their respective purified (HA-MX and HA-TPH) and functionalized humic acids with $CH_3I$ (MI), $C_7H_7Br$ (BBr), and $C_5H_{11}Br$ (PBr).

| Humic Acids | %C (±SD) | | %H (±SD) | | % N (±SD) | |
|---|---|---|---|---|---|---|
| | **-MX-** | **-TPH-** | **-MX-** | **-TPH-** | **-MX-** | **-TPH-** |
| HA sources | 13.9 (±0.2) [g] | 40.9 (±0.2) [f] | 1.31 (±0.001) [g] | 3.76 (±0.11) [d] | 0.22 (±0.03) [g] | 0.75 (±0.01) [f] |
| Pure HAs | 57.5 (±0.5) [bcd] | 46.5 (±0.3) [e] | 2.63 (±0.11) [f] | 3.14 (±0.11) [e] | 1.10 (±0.01) [b] | 1.01 (±0.01) [e] |
| | Functionalized humic acids: | | | | | |
| MI | 62.3 (±0.11) [d] | 57.3 (±0.2) [cd] | 3.79 (±0.01) [f] | 4.58 (±0.11) [c] | 1.41 (±0.02) [d] | 1.57 (±0.01) [b] |
| BBr | 58.9 (±0.3) [b] | 66.1 (±0.5) [bcd] | 3.12 (±0.01) [ef] | 5.53 (±0.06) [a] | 1.14 (±0.01) [c] | 1.26 (±0.02) [a] |
| PBr | 59.2 (±0.2) [bc] | 58.2 (±0.4) [a] | 2.97 (±0.08) [ef] | 6.05 (±0.10) [b] | 1.25 (±0.01) [d] | 1.69 (±0.01) [c] |

One-way ANOVA. By element, means that do not share a letter are significantly different. Tukey's test at 95% confidence. (-MX-) indicates Mexican origin; (-TPH-) indicates the origin of humic acids (Humin Tech Pow Humus WSG).

The H/C ratio lower than 1 (Table 1) and the low E4/E6 ratios ($(E4/E6)_{LEO-MX} = 4.7 \pm 0.4$; $(E4/E6)_{HTPH} = 6.6 \pm 0.6$) determined for the two HA sources indicate that the available HAs have a high level of humification. The ash content in the LEO-MX and HTPH humic substances was $57.3 \pm 0.4\%$ and $31 \pm 0.1\%$, respectively, and $1.2 \pm 0.3\%$ and $1.6 \pm 0.2\%$ in their respective purified humic acids.

According to Droussi et al. [26], a low content of elemental carbon is an indicator of stable humic substances, that is, substances that have been subjected to a long humification process. Based on this parameter, of the two types of HA studied here, the HTPH are the more aged HAs (Table 1). These HAs incorporated the higher percentage of carbon atoms ($42.1 \pm 0.5\%$) in their reaction with BBr, and H and N with PBr. The concentration of these elements augmented by $92.8 \pm 5.1\%$ and $67.3 \pm 1\%$, respectively, in the HA-TPH reaction with the pentyl bromide.

*3.2. Solubility, pH, and Surface Tension*

The two sources of HA studied showed similar physicochemical properties concerning the surface tension and solubility (Table 2). After the HA extraction and purification, only the HA-TPH properties changed; it became insoluble at a pH of 9, and the surface tension diminished by 6.8%. Solubilization of these acids was achieved at a pH of 11.5. The critical micelle concentration of the pure HA-MX and HA-TPH was 50 and 70 mg $L^{-1}$, respectively.

The chemical modification of HAs changed their surfactant properties. While the MI decreased the surface tension of HA, the other two alkyl halides produced the opposite effect. The magnitude of the surface tension changes presented significant differences for a $p(\text{F}) < 0.05$ (Table 2).

**Table 2.** Physicochemical properties of the two humic substances (LEO-MX and HTPH) and their respective pure (HA-MX and HA-TPH) and functionalized humic acids with $CH_3I$ (MI), $C_7H_7Br$ (BBr), and $C_5H_{11}Br$ (PBr).

| Humic Acids | Surface Tension (mN m$^{-1}$) | | pH of HA in Solution | | NaOH (M) Required to Solubilize the Tested HA | |
|---|---|---|---|---|---|---|
| | **-MX-** | **-TPH-** | **-MX-** | **-TPH-** | **-MX-** | **-TPH-** |
| HA sources | 64.7 (±0.2) [bc] | 64.7 (±0.2) [bc] | 9.3 (±0) [e] | 9.2 (±0) [e] | 0.0 (±0) | 0.0 (±0) |
| Pure HAs | 64.8 (±0.7) [bc] | 60.3 (±1.1) [de] | 9 (±0) [f] | 11.5 (±0) [d] | 0.0 (±0) | 0.016 (±0) |
| | | | Functionalized humid acids: | | | |
| MI | 62.4 (±1.1) [cd] | 58 (±2) [e] | 12.1 (±0) [bc] | 12.0 (±0) [c] | 0.02 (±0) | 0.013 (±0) |
| BBr | 67.3 (±0.1) [ab] | 63.2 (±0.1) [cd] | 12.1 (±0) [bc] | 12.2 (±0) [ab] | 0.02 (±0) | 0.013 (±0) |
| PBr | 68.7 (±0.1) [a] | 62 (±0.2) [cd] | 12.3 (±0.1) [a] | 12.1 (±0.1) [bc] | 0.02 (±0) | 0.016 (±0) |

One-way ANOVA. By physicochemical property, means that do not share a letter are significantly different. Tukey's test at 95% confidence. (-MX-) indicates Mexican origin; (-TPH-) indicates the origin of humic acids (Humin Tech Pow Humus WSG).

Humic acids are the soluble fraction of humic substances [27], although it has also been reported that HA solubility depends on the alkalinity of the solutions used in the extraction process. Thus, the higher the pH, the greater the molecular weight of the extracted HAs and the lower their solubility in water [18]. The loss of solubility of the pure HA-TPH could be due to the elimination of carboxylic acids during the purification process as reported by Maryganova et al. [28].

Surface tension is another variable affected by the HA structure, and the pH and concentration at which the measurement is performed. Klavins and Purmalis [27] found an inverse relationship between the humification degrees of HA and the surface tension of their solutions. Ramírez-Cutiño et al. [29] reported a diminution of the surface tension when the HA concentration was increased, and the pH diminished in the 3 to 4 range. Concerning the surface tension of the HAs studied here, the relatively low drop in their surface tension relative to the control (deionized water) could be attributed to their relatively high degree of humification, which is indicated by their low H/C ratio (<0.1) and an E4/E6 ratio lower than 10 [30,31].

### 3.3. FTIR and $^{13}$C-NMR Spectra of the Tested Humic Acids

The FTIR vibrations registered in the LEO-MX and their corresponding purified and functionalized HAs are shown in Figure 1A. Vibrations were attributed to the following groups: (3400 cm$^{-1}$) -OH stretch presented in groups of alcohols, phenols, and carboxylic compounds; (1715 cm$^{-1}$) stretching vibrations C=O attributed to the presence of carboxyl groups and ketones; (1590–1600 cm$^{-1}$) vibrations produced by the DC voltage in aromatic rings, asymmetric tension of COO, C=O, or voltage C=C conjugated with carbonyl groups or other double bonds (ethers and esters); (1390 cm$^{-1}$) flexion of the -OH in alcohols, -OH deformation of the COOH in carboxylic acids and phenols, or -CH deformations in -CH2 and -CH3 groups; a band centered around 1200 cm$^{-1}$ represents a C-O stretch or -OH stretch present in phenols, carboxylic acids, esters, and ethers.

The highest number of bands was observed in the LEO-MX spectra. However, those located at both ends (in 3690, 3620, 528, and 467 cm$^{-1}$) are associated with aluminosilicates, the main components of the HA ashes [32,33]. Additional peaks in the LEO-MX spectrum come from vibrations of the following chemical structures: -CO stretching in alcohols (1100 cm$^{-1}$), the -S=O in sulfoxides (1030 cm$^{-1}$), and the -OH stretch in carbohydrate (1010 cm$^{-1}$); the deformation -OH and doubling -C=O in the carboxylic acids is located in the 910 and 536 cm$^{-1}$ bands [34].

Signals of aluminosilicates disappear in the pure HA-MX, and those remaining correspond to the following groups: aliphatic (-CH2, -CH3, 2940–2840 cm$^{-1}$), carboxylic (C=O-, 1650–1800 cm$^{-1}$), methyl (-CH, 1370–1450 cm$^{-1}$), and phenols (-CO, 1150–1300 cm$^{-1}$). These signals were additionally increased with the chemical modification of the HA-MX as shown in Figure 1A. Of these signals, those of the HA-MX modified with benzyl bromide (BBr) stand out.

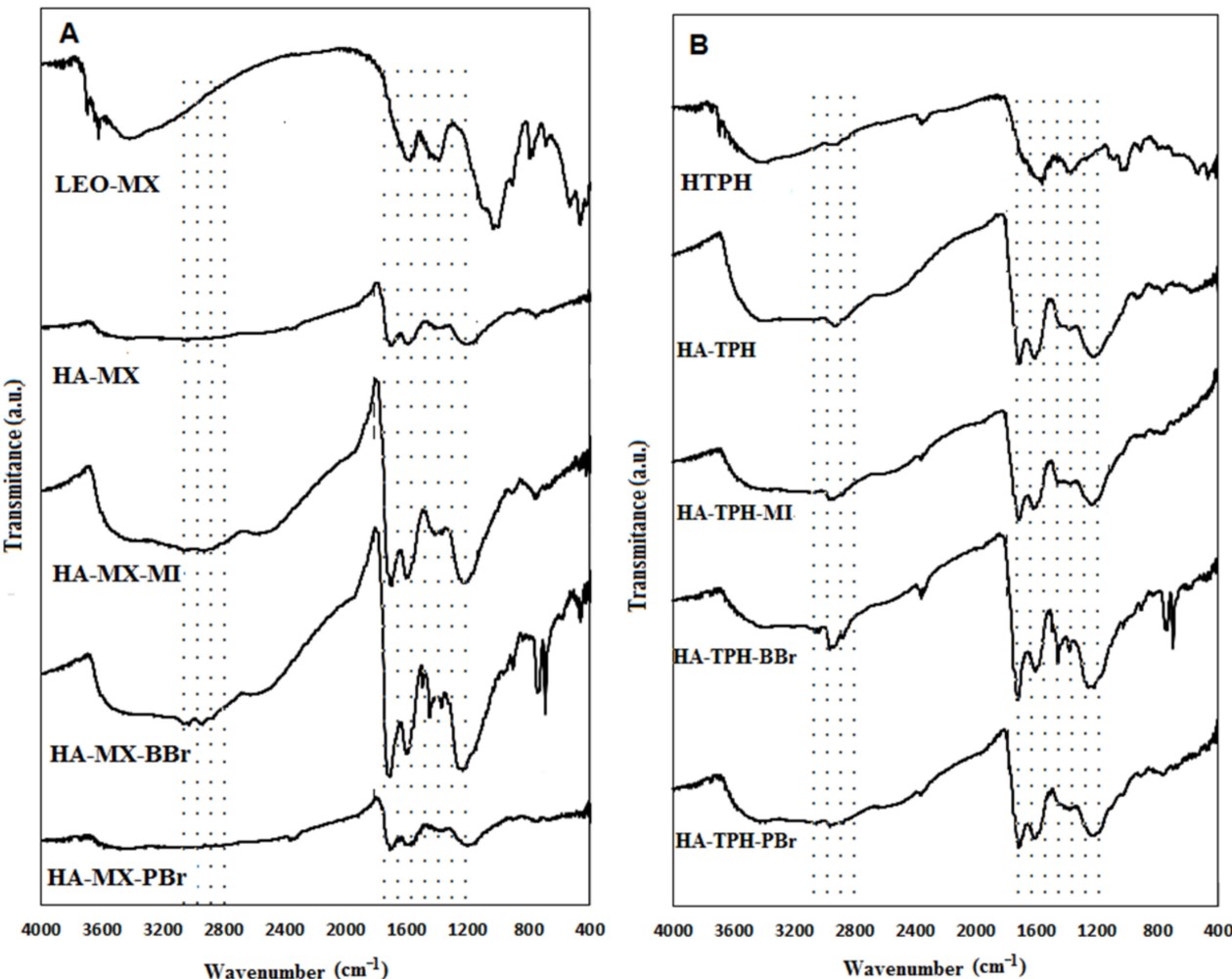

**Figure 1.** FTIR spectra of humic acids. (**A**) Mexican leonardite (LEO-MX) and respective pure (HA-MX) and functionalized (HA-MX-MI, HA-MX-BBr, HA-MX-PBr) humic acids. (**B**) Humin Tech Pow Humus (HTPH) and respective pure (HA-TPH) and functionalized (HA-TPH-MI, HA-TPH-BBr, HA-TPH-PBr) humic acids.

Aluminosilicate impurities (3300–2300 cm$^{-1}$ range) were detected at a lower concentration in the commercial HTPH (Figure 1B). Unlike the LEO-MX spectrum, an additional band was detected at 2920 cm$^{-1}$, which corresponds to the aliphatic group.

The HA functionalization with MI generated more intense signals in the pure HA-MX than in the HA-TPH, specifically in the range 3400–900 cm$^{-1}$. Of the HA reactions with PBr and BBr, those with PBr generated signals of greater intensity in both HA-TPH and HA-MX humic acids. In these spectra, the functional group of aliphatics (2840–2940 cm$^{-1}$) stood out in the HA-TPH and the aromatic signals (1715–1220 cm$^{-1}$) in the HA-MX. Predominance of these functional groups, for each case, was verified by $^{13}$C-NMR analysis as shown in Figure 2A. The aliphatic, aromatic, and carboxyl groups can be identified in the regions 0–50, 108–165, and 165–190 ppm, respectively [35].

In the NMR spectra of the HA-MX, the aromatic group is centered at 128 ppm, the carboxyl at 164 ppm, and aliphatics at 22 ppm. In the case of the HA-TPH NMR spectra, the aliphatic group is centered at 22 ppm and the aromatics at 135 ppm. The alkyl halide MI gave rise to compounds derived from aldehydes and ketones, while the other two halides, PBr and BBr, generated aliphatic compounds with carbons substituted by oxygen or nitrogen (C-O, C=O or C=N). The signals in the 10–11 ppm region were identified as carbons with sp$^3$ hybridization. In the HA-TPH, the carbonyl functional group is scarce in comparison to the corresponding one observed in the HA-MX (Figure 2B).

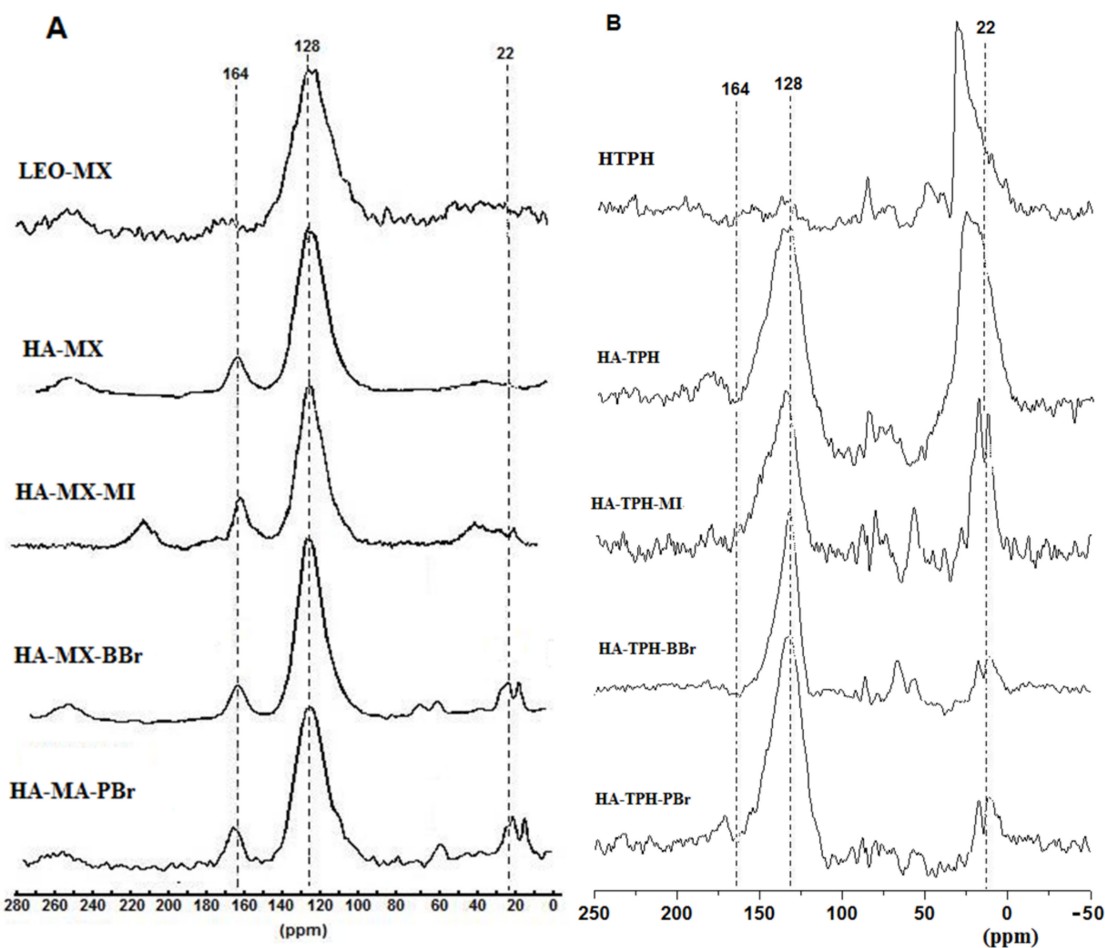

**Figure 2.** $^{13}$C-NMR spectra of humic acids. (**A**). Mexican leonardite and respective pure (HA-MX) and functionalized (HA-MX-MI, HA-MX-BBr, HA-MX-PBr) humic acids. (**B**). Humin Tech Pow Humus (HTPH) and respective pure (HA-TPH) and functionalized (HA-TPH-MI, HA-TPH-BBr, HA-TPH-PBr) humic acids.

The main functional groups that conform to the complex HA molecular structure were successfully identified through the FTIR technology. However, the small number of peaks in the spectra is due to the large number of vibrational signals, which can be canceled [34]. The vibrational signals that are commonly seen are those of the predominating functional groups found in high concentrations. Comparing the FTIR spectra of the non-treated and purified HA-MX with the corresponding ones of the HA-TPH, their notable similarity can be distinguished. These results are consistent with those reported by Fooken and Liebezeit [32], who pointed out that the greatest difference between the FTIR spectra of HAs of different origins lies in the intensity of the peaks rather than in the location of bands in the spectrum. Observing the $^{13}$C-NMR spectra, the chemical shifts of -CH$_3$ groups in methyl ethers and esters vary over a broad range depending on their structural position and their chemical environment.

### 3.4. Sorption of PAHs in Selected Humic Acids

The sorption of selected PAHs in the different HAs studied (source, pure, and functionalized HAs) generated the FTIR spectra presented in Figure 3. With some variations in the number of peaks, PAHs changed the spectrum of HA. They give rise to a characteristic spectrum for the new molecular structure formed by the HAs and PAHs (HAx-PAHx), and peak intensity changed with the type of HA used as sorbent. The effect of the PAH type on the FTIR spectra of the HA-TPH functionalized with PBr (HA-TPH-PBr) is shown in Figure 3A. Figure 3B shows the changes produced by the anthracene on the FTIR spec-

tra of the pure HA-TPH and functionalized HA-TPH (HA-TPH-MI, HA-TPH-BBr, and HA-TPH-PBr), after the sorption of these PAHs in the different HAs. The increased HA spectrum signals with PAH sorption are assigned to: 3620 cm$^{-1}$ (H-bonded OH groups of alcohols, phenols, and organic acids, as well as H bonded in N-H groups), 3400 cm$^{-1}$ ($\nu$(0H) stretching motion of carboxylic and alcoholic groups), 2500 cm$^{-1}$ (aliphatic C-H stretch), 1417 cm$^{-1}$ ($\delta$(CH2) motion of aliphatic groups), and 1018 cm$^{-1}$ (C-O stretching observed in carbohydrates and $\nu$ (C-C) skeletal vibration of aliphatic groups) (Figure 3A) [36,37]. The spectrum signals in which a diminution was observed are as follows: 2926 cm$^{-1}$ (C-H stretching of alkyl structures), 1716 cm$^{-1}$ (C=O of COOH, C=O stretch of ketonic C=O), 1601 cm$^{-1}$ (aromatic C=C, C=O in amide (I), ketone and quinone groups), and 1207 cm$^{-1}$ (amides and ethers). The PAHs sorbed in the LEO-MX and HTPH humic substances did not change their respective FTIR spectra; however, the HTPH spectrum presented peaks of slightly greater intensity (Figure S1 in Supplementary Materials).

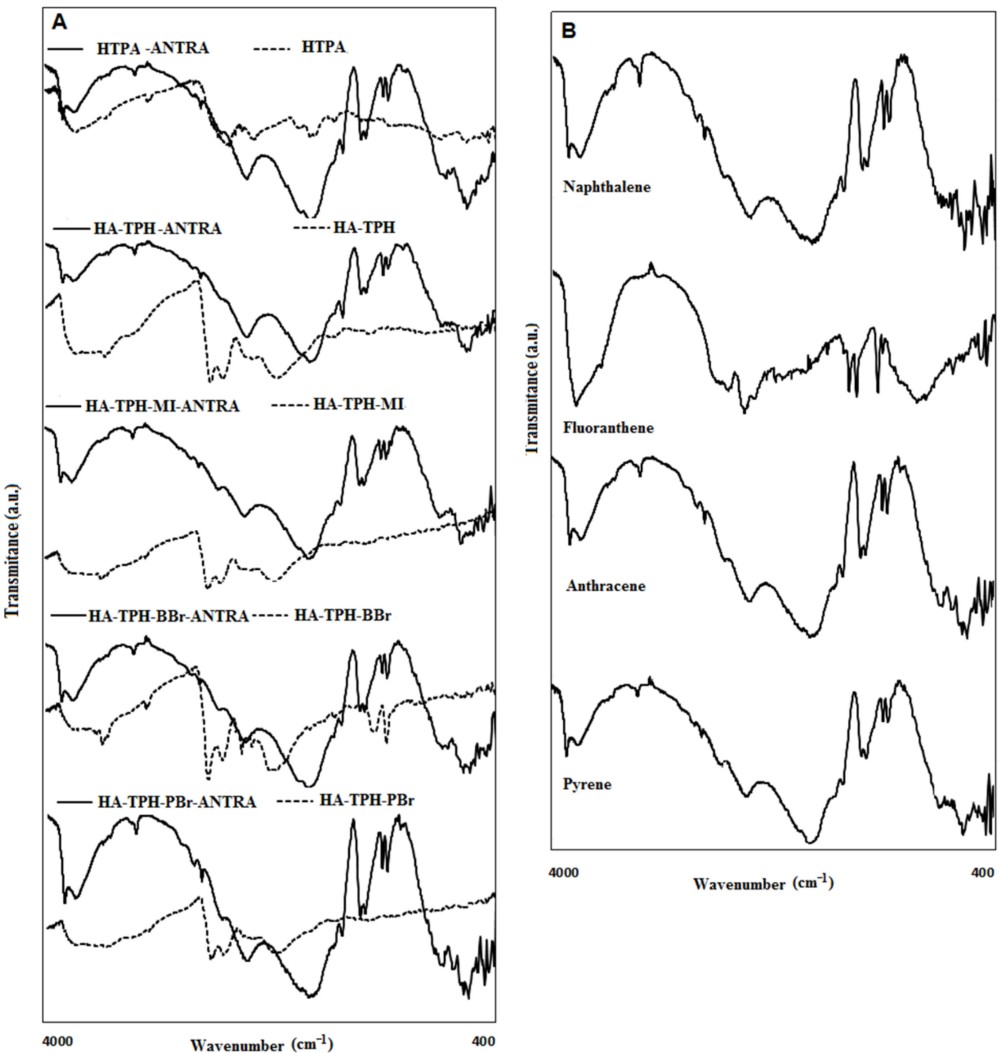

**Figure 3.** (**A**) FTIR spectra of humic substances (HTPH, HA-TPH, and functionalized humic acids (HA-TPH-MI, HA-TPH-BBr, HA-TPH-PBr)) with anthracene (ANTRA) sorbed. (**B**) FTIR spectra of pure and functionalized humic acids with pentyl bromide (PBr) obtained from the Humin Tech Pow Humus (HTPH) and with the following PAHs sorbed: naphthalene, fluoranthene, anthracene, and pyrene.

According to the literature, PAHs can be entrapped in the aliphatic and carboxylic functional groups [38]. The NMR signals produced by these types of functional groups showed a marked decrease in the NMR spectrum of the HTPH and pure or functionalized

HA-TPH humic acids. This assumption is supported by the appearance of new stable signals in the 2000–1793 $cm^{-1}$ region of the spectrum (Figure 3). This represents unusable replacement aromatic rings with substituents such as C=O, $NO_2$, and CN [39,40]. The bonds formed are classified as strongly conjugated and emit vibrations out of the CH plane of deformation, in the region 829–658 $cm^{-1}$. These vibrations can be used to identify the position and number of substituents in the aromatic ring. Because of this, the HA-TPH-PBr showed the highest signal of the functional group of aliphatic hydrocarbons, and this was the HA selected for the purposes of PAH sorption in the HA immobilized in zeolite.

### 3.5. Immobilization in Zeolite of Humic Acids

#### 3.5.1. Characterization of Zeolite

Sigma-Aldrich zeolite (<45 μm) was reduced to an average diameter of 170 ± 43 nm in 12 h, as shown in Figure 4. The grinding process was performed for 96 h; however, the particle size did not change significantly after the twelfth hour of grinding. Oxygen had the highest concentration in the zeolite (47.6 ± 6.8%) and carbon the lowest (1.83 ± 0.79%). Silica, aluminum, and sodium were calculated at percentages of 20.1 ± 0.97%, 15.9 ± 0.84%, and 14.5 ± 1.0%, respectively (Table S1 in Supplementary Materials). The color-coded (SEM) elemental composition of the zeolite is shown in Figure 4D.

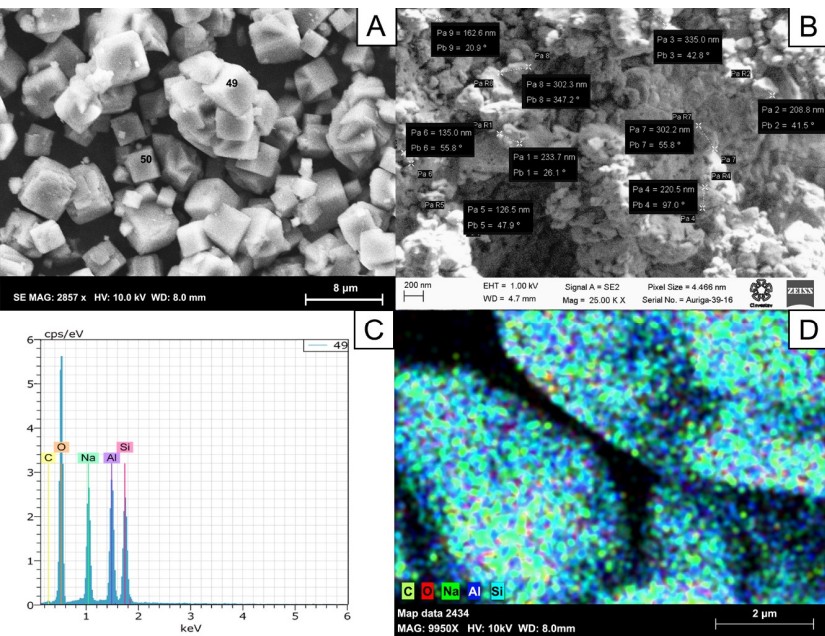

**Figure 4.** (**A**) Zeolite from Sigma-Aldrich (<45 μm). (**B**) Zeolite ground for 96 h in a ball mill (170 ± 43 nm). (**C**) Elemental analysis of zeolite. (**D**) Elemental analysis of zeolite by high-resolution SEM.

The size reduction of the zeolite increased its specific surface area from 53 ± 10 to 228 ± 23 $m^2 g^{-1}$. After the HA sorption in the nano-zeolite, an increment on the surface area to 4172 ± 10 $m^2 g^{-1}$ was calculated. Such a surface increment should allow an improved sorption of PAH.

#### 3.5.2. Sorption Isotherms of Humic Acids in Zeolite of Two Particle Sizes

This test was performed with the following functionalized HAs: HA-TPH-PBr and HA-MX-PBr. The zeolite particle size and the concentration and pH of the humic acid solutions showed a significant effect with a $p(F) < 0.05$ (Figure 5). Zeolite with nanometric size was more effective in adsorbing the HA in solution at a pH of 0.5. The sorption process was described by the Freundlich model when using nano-zeolite and by a linear model in the case of micro-zeolite. The adjustment of the experimental data in different sorption models allows us to assume that nano-zeolite is better suited to immobilizing the selected

functionalized HA. The effect of the HA type on the sorption process was only observed in HA solutions of concentrations greater than 50 mg L$^{-1}$. The pH of the HA solutions and the zeolite particle size showed important effects when HA solutions of low concentration (10 mg L$^{-1}$) were used.

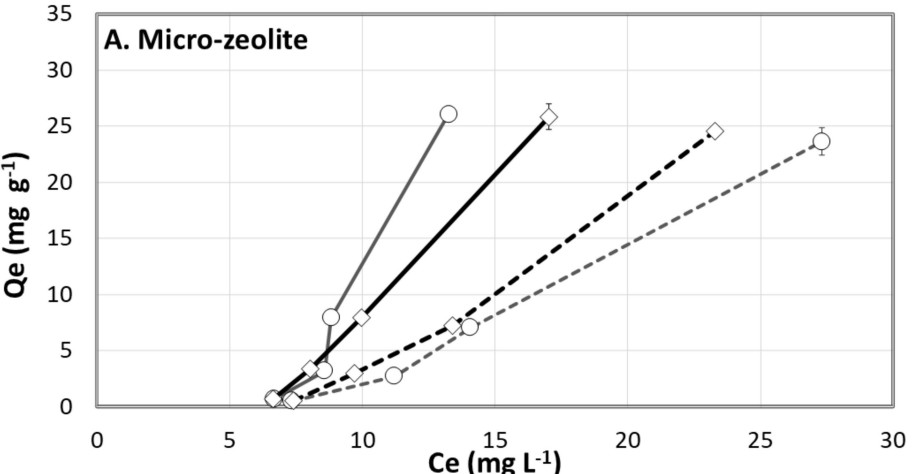

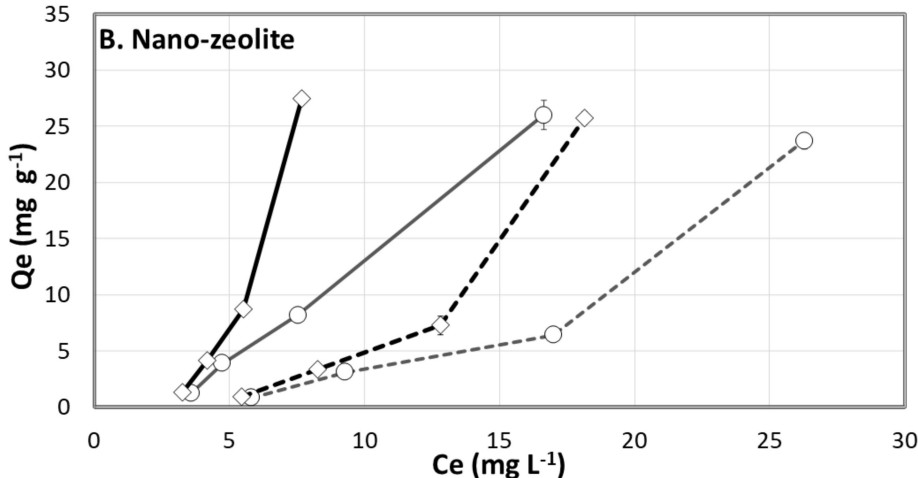

**Figure 5.** pH effect on the adsorption isotherms of the HA-TPH (diamonds) and HA-MX (circles) in zeolite particles of two sizes: (**A**) micro, (**B**) nano. The continuous and interrupted lines are the prediction of experimental data using the Freundlich model for pH values of 0.5 and 3.5, respectively.

Sorption in zeolite of the HA-TPH-PBr and HA-MX-PBr humic acids showed a different behavior when the zeolite size was changed from micro (<45 μm) to nanometric size (<170 ± 43 nm). HA sorption in the micro-zeolite followed a model that is closer to Henry's linear isotherms [41]. Linear models that intersect with the positive axis of the abscissa have been previously reported by Leone et al. [42], and the positive displacement from the origin has been attributed to a rapid adsorption of the sorbent in the solid support.

The Freundlich isotherms successfully correlated the HA sorption in the nano-zeolite with a $R^2$ > 0.93 and a $p(F)$ < 0.05 (Figure 5). The good fit of this model with *n* values lower than 1 could be due to the heterogeneity of the zeolite surface after the grinding process. It is known that the HA surfactant properties and the negative charge of the zeolite produce weak links between these two components [43]. However, in the present study, where functionalized humic acids were used and added at high concentrations (10–150 mg L$^{-1}$), the percentages of 63–95% for a pH of 0.5 are assumed to be adequate for its application in the treatment of wastewater where HAs have been found at a lower

concentration (5–10 mg L$^{-1}$). High percentages of HA sorption in nano-zeolite (83.16%) have been reported by Tashauoei et al. [44]. They evaluated the sorption capacity of nano-A-zeolite using HA from Sigma-Aldrich at concentrations of 10 mg L$^{-1}$ and pH = 3. In the present study, the nano-zeolite allowed HA sorption at percentages higher than 90%; thus, it seems to be a good alternative for PAH removal. The solid/liquid (S(mg)/L(g)) ratio in the 3.5 to 5 range is considered suitable for HA adsorption in nano-zeolite [8]. In the present study, the S/L ratio was 5, which could explain the results obtained.

Studies on the capacity of natural zeolites of different origins to sorb humic acids are extensive. The availability of this type of information allows a better use of these resources in solving environmental pollution problems. In the present work, the mass of HAs sorbed per gram of zeolite was in the range of 4 to 27 mg g$^{-1}$ using an HA solution of an initial concentration of 30–150 mg L$^{-1}$. Values of this magnitude have already been reported by Zhan et al. [45] but using a 30 mg L$^{-1}$ humic acid solution and zeolites (500 mg L$^{-1}$) treated with HTAB at a concentration of 50 mmol L$^{-1}$. Capasso et al. [46] reported an HA sorption capacity of 8.5 mg g$^{-1}$ using Italian natural zeolites at a concentration of 790 $\pm$ 76 mg L$^{-1}$. This value is lower than that obtained in our work, despite the high concentration of HA used. Wang et al. [22] reported a sorption capacity of 37 mg of humic acids in a natural zeolite of Australian origin. These authors used a humic acid solution of 30 mg L$^{-1}$ and pH = 5. The particular characteristics of this zeolite give it a greater capacity to adsorb humic acids.

*3.6. Sorption of PAHs in Functionalized HA Immobilized in Zeolite*

This test was performed with the HA-TPH functionalized with the PBr (HA-TPH-PBr) and the following PAHs: fluoranthene, anthracene, and pyrene. The sorption percentage of the PAHs in the micro- and nano-zeolites, alone or with the immobilized HA-TPH-PBr, is shown in Figure 6 and the mass of hydrocarbons adsorbed per unit mass of sorbent in Table 3. The HA-TPH-PBr and the micro-zeolite sorbed the three PAHs with an average of 28 $\pm$ 5%, and by 56.4 $\pm$ 13% when the HA-TPH-PBr had been previously immobilized in the micro-zeolite. The mean PAH sorption was increased by 16.7% when nano-zeolite was used. Anthracene and pyrene, the PAHs with the greater ring number, were sorbed by the nano-zeolite at percentages of 82 $\pm$ 1% and 84.9 $\pm$ 1%, respectively. The corresponding remaining PAH concentration was 5.3 $\pm$ 0.3 and 4.5 $\pm$ 0.3 mg L$^{-1}$, from an initial concentration of 30 mg L$^{-1}$.

The capacity of the hybrid sorbents evaluated here to adsorb the fluoranthene, anthracene, and pyrene PAHs presented values within the maximum values reported in the literature and even higher (Table 3). Wołowiec et al. [47], who tested the capacity of zeolites, organo-zeolites, and synthetic zeolites (organo-Na-X) to adsorb anthracene (initial concentration of 20 mg L$^{-1}$), reported values within the range of 0.08 and 0.14 mg g$^{-1}$, the highest values of which correspond to the organo-zeolites and organo-synthetic zeolites. The capacity of zeolites and zeolites modified with DDAB, CPC, and HDTMA presented an adsorbing capacity of 9.7–10, 9.7–9.9, and 4.7–4.9 μg g−1, to adsorb fluoranthene, anthracene, and pyrene, respectively [8].

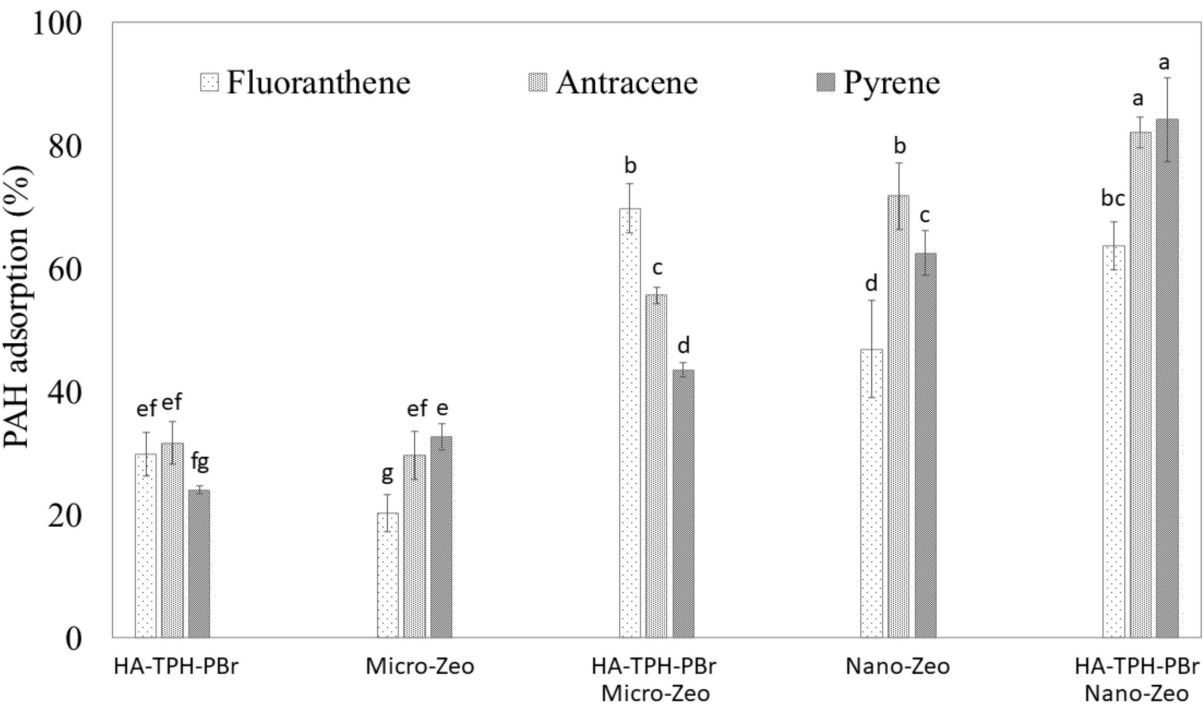

**Figure 6.** PAH sorption in functionalized humic acids (HA-TPH-PBr), micro-zeolite (Micro-Zeo), nano-zeolite (Nano-Zeo), and two hybrid sorbents ((HA-TPH-PBr)-(Micro-Zeo) and (HA-TPH-PBr)-(Nano-Zeo)). Means that do not share a letter are significantly different. Tukey's method and 95% confidence.

**Table 3.** Sorption of fluoranthene, anthracene, and pyrene in the following sorbents: humic acids HA-TPH modified with PBr (HA-TPH-PBr), micro-zeolite (Micro-Zeo), nano-zeolite (Nano-Zeo), and in hybrid supports (HA-TPH-PBr)-Micro-Zeo and (HA-TPH-PBr)-Nano-Zeo.

| Sorbent | Fluoranthene (mg g$^{-1}$) | Anthracene (mg g$^{-1}$) | Pyrene (mg g$^{-1}$) |
|---|---|---|---|
| HA-TPH-PBr | 1.9 ± 0.3 [ef] | 2.0 ± 0.2 [ef] | 1.6 ± 0.1 [fg] |
| Micro-Zeo | 1.3 ± 0.3 [g] | 1.9 ± 0.3 [ef] | 2.0 ± 0.2 [e] |
| (HA-TPH-PBr)-(Micro-Zeo) | 4.4 ± 0.04 [b] | 3.5 ± 0.1 [c] | 2.7 ± 0.1 [d] |
| Nano-Zeo | 2.93 ± 0.5 [d] | 4.5 ± 0.1 [b] | 3.8 ± 0.2 [c] |
| (HA-TPH-PBr)-(Nano-Zeo) | 4.0 ± 0.2 [bc] | 5.13 ± 0.1 [a] | 5.3 ± 0.1 [a] |

Two-way ANOVA. Means that do not share a letter are significantly different. Tukey's test at 95% confidence.

The removal of PAHs by natural zeolites has been reported at percentages of 3–30% [8]. However, higher removal percentages have been reported when surfactant-modified zeolites were used. Fluoranthene, anthracene, and pyrene, for example, were removed at percentages above 93% when using clinoptilolite modified with CPC, DDAB, and HDTMA. The initial concentration of PAHs was 0.1, 0.05, and 0.1 mg L$^{-1}$, respectively, and the ratio between the solid sorbent and the PAH solution was 1:100 [8]. The effect of the particle diameter on the removal of PAH by sorption with organo-zeolite (zeolite treated with stearyldimethylbenzylammoniumchloride (SDBAC)) has been reported by Lemić et al. [11]. These authors found that the organo-zeolite with smaller diameter (0–0.4 mm) favored the total removal of fluoranthene and pyrene added to an aqueous solution at a concentration of 0.01 mg L$^{-1}$. The PAH removal capacity of this organo-zeolite was 0.68 and 0.63 mg g$^{-1}$, respectively. The removal of anthracene (200 pg) with natural zeolite was reported by Manni et al. [48]. When 10 g of zeolite was suspended in 150 mL of anthracene solution, 99.7% of PAHs were removed. Plaza et al. [38] reported the removal of pyrene with HA at a concentration of 25 mg L$^{-1}$. The sorption capacity of HA was 3.4 mg g$^{-1}$ for an initial concentration of pyrene of 0.11 mg L$^{-1}$. The removal percentages reported by the cited

authors are high; however, it is important to point out the low initial concentration of the PAHs, which was below their solubility in water.

Zeolites and HAs are substances that are freely found in nature and have been reported to have the ability to trap a wide variety of contaminants [43]. The chemical treatment of these substances is a practice commonly used to increase their capacity as sorbents, as well as to extend their application to remove a greater variety of polar and non-polar pollutants [49]. The type of modification will depend on the type of contaminants to be removed. In the present work, the trapping of hydrocarbons with zeolites reduced to nanoparticle diameters and the functionalization of humic acids to trap aromatic-type compounds allowed the elimination of aromatic hydrocarbons with three and four rings. The lower percentage of sorption in nanoparticles of fluoranthene has been attributed to their angled molecular structure, which limits their linkage [50]. The crystallinity of water when submerging the solid support HA-TPH-PBr-PAHs allows us to assume the irreversibility of the sorption process not only of HA but also of PAHs.

## 4. Conclusions

Hybrid zeolite/HA sorbents with a great capacity to remove high molecular weight PAHs were synthesized. The functionalization of the HAs used in this study increased their non-polarity, as well as the formation of aliphatic groups. This type of structure favored the sorption of hydrophobic compounds such as PAHs. Zeolites are microporous materials with high sorbent capacity, and a decrease in their particle size extends their specific surface area and consequently sorbent capacity. It is assumed that the immobilization of HA in hybrid supports of a large specific surface, in addition to favoring the entrapment of PAHs, and its immiscible solid structure, presents the adequate characteristics for its separation from aqueous effluents by physical methods. The complex structure of HAs gives them extraordinary properties, which are being explored for their application in solving problems in the health sector such as environmental pollution. The availability in nature of both zeolites and humic acids would make it possible to synthesize these hybrid supports on a larger scale for their application in the remediation of contaminated systems. Humic acids are a source of nutrients, so the microbial flora associated with these materials could be used for the degradation of pollutants and could later be recycled for reuse in the treatment of contaminated aqueous systems.

**Supplementary Materials:** The following are available online at https://www.mdpi.com/article/10.3390/su131810391/s1, Figure S1: FTIR spectra of functionalized humic acids (HA-MX-PBr and HA-TPH-PBr)) with (A) pyrene (PYR) and (B) anthracene (ANTRA) sorbed., Table S1: Elemental chemical composition of zeolite determined by high resolution SEM.

**Author Contributions:** G.R.-M. and J.B.-C.: substantial contributions to the conception and design of the work, acquisition, analysis, and interpretation of experimental data. L.V.-C.: support, discussion, analysis, and interpretation and graphing of FTIR and NMR spectra. O.S.-F.: support and discussion, optical characterization of the zeolite and humic acids, and revision of the manuscript content. C.G.-D.: advice on the chemical modification of humic substances. All authors have read and agreed to the published version of the manuscript.

**Funding:** Research supported by Conacyt through projects CB-2010-156837 and INFRA-2012-01-188339.

**Institutional Review Board Statement:** Not applicable.

**Informed Consent Statement:** Not applicable.

**Data Availability Statement:** The experimental data are already included in the article in the form of graphs and tables. Additional information will be provided upon request.

**Acknowledgments:** The authors would like to thank Juan Méndez Vivar from UAM-I, as well as to María de Lourdes Rojas Morales (LANSE Cinvestav-IPN) and Josué Romero Ibarra (LANE-HRSEM Cinvestav-IPN) for their technical and steadfast help.

**Conflicts of Interest:** The authors declare no conflict of interest.

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
