# Peer review of "Polycyclic Aromatic Hydrocarbon Sorption by Functionalized Humic Acids Immobilized in Micro- and Nano-Zeolites"

_sustainability, doi:10.3390/su131810391_

Round 1

Reviewer 1 Report

The article is generally well written, however before being considered for publication the following major issues should be considered:

  1. The authors use both the terms of adsorption and absorption, although I think that they refer from all points of view to adsorption. Considering the different definitions and actions of these processes, the  clarification of the process studied should be made all over the text.
  2. It is not clear if the studied sorbents can be synthesized and used at large scale for wastewater treatment

  3. There are only few sorption tests to support the affirmation:  “....in addition to favoring the entrapment of PAHs, will allow their separation in water treatment applications” only by calculating the sorption removal

  4.  

     It is not shown the sorption capacity obtained with the final synthesized material. Only a figure is presented and this shows only the removal efficiencies. 

  5.  

    The pH values of 0.5 si 3.5 were applied without having as support a study, a previous study results were used (Tashauoei, H.R.; Movahedian Attar, H.; Amin, M.M.; Kamali, M.; Nikaeen, M.; Vahid Dastjerdi, M., Removal of cadmium and  humic acid from aqueous solutions using surface modified nanozeolite A. Int. J. Environ. Sci. Tech. 2010, 7 (3), 497-508  [https://doi.org/10.1007/BF03326159]

  6.  

    No study is presented to compare the sorption capacity and results with other data and studies available in the scientific literature
  7. No tests are available with real wastewater effluents

  8. No sustainability considerations are made with reference to the materials or process studied, although this is the specific target of this journal
  9. There are missing the authors details, institutions, e-mails, etc and this is a problem also for assuming the authorship of this study.

Author Response

The authors appreciate the suggestions and comments made by the reviewers. They allowed us to substantially improve the presentation of the proposed manuscript for its publication in the Sustainability Journal.

The specific questions were answered individually, in order of appearance in the document received with annotations. The corrected manuscript was submitted for review of English grammar.

Reviewer: 1

The article is generally well written, however before being considered for publication the following major issues should be considered:

  1. The authors use both the terms of adsorption and absorption, although I think that they refer from all points of view to adsorption. Considering the different definitions and actions of these processes, the clarification of the process studied should be made all over the text.

The authors thank reviewer 1 for their observation. The pertinent corrections were made in the manuscript.

Some articles refer to the removal of pollutants by absorption (Hedayati & Li 2020). We agree that this term is limited in the problem discussed in the paper.

In the literature on the removal of HA with zeolites, or the removal of PAH with HA and zeolites, reference is made to the sorption of one material into another, as well as to the adsorption isotherms (Wang et al., 2016). Sorption is a general term that covers absorption, adsorption, and ion exchange (Chianese et al., 2020, Lamichhane et al., 2016). Absorption refers to the entry of one material into a porous material. Adsorption is the interaction between two materials, but at surface level. By definition, adsorption isotherms are reported, however, the characteristics of the adsorbate and adsorbent materials determine the type of interaction between them.

“Sorption process, a generic term covering the processes of adsorption, adsorption and ionic exchange, is one of the most effective methods for wastewater treatment, because of its relatively high efficiency, low-cost, and simplicity”

Hedayati, M.S.; Li, L.Y. Removal of polycyclic aromatic hydrocarbons from aqueous media using modified clinoptilolite. J. Environ. Manage. 2020, 273, 111113 [https://doi.org/10.1016/j.jenvman.2020.111113]

Chianese S. ; Fenti A. ; Iovino P. ; Musmarra D.; Salvestrini S. Sorption of Organic Pollutants by Humic Acids: A Review. Molecules 2020, 25(4), 918 [doi:10.3390/molecules25040918]

Lamichhane, S.; Krishna, K.C.B.; Sarukkalige, R. Polycyclic aromatic hydrocarbons (PAHs) removal by sorption: A review. Chemosphere 2016, 148, 336-353. [https://doi.org/10.1016/j.chemosphere.2016.01.036]

Wang Z., Li Y., Guo P., Meng W. Analyzing the Adaption of Different Adsorption Models for Describing the Shale Gas Adsorption Law. Chem. Eng. Technol. 2016, 39, No. 10, 1921–1932

  1. It is not clear if the studied sorbents can be synthesized and used at large scale for wastewater treatment

Zeolites and humic substances are materials that are considered readily available. Zeolites are materials that are found in nature and that can even be synthesized easily and according to the desired characteristics (Hardi et al., 2020; Dusselier and Davis, 2018). The humic substances used in the research carried out are offered commercially as fertilizers. The synthesis of hybrid zeolites was carried out at the laboratory level, however, considering that they are materials available in nature, their production on a larger scale should be possible.

The following phrase was included in the conclusion (lines: 516-521):

“The availability in nature of both zeolites and humic acids makes it possible to synthesize these hybrid supports on a larger scale for their application in the remediation of contam-inated systems. Humic acids are a source of nutrients, so the microbial flora associated with these materials could be used for the degradation of pollutants and could later be re-cycled for reuse in the treatment of contaminated aqueous systems”

Hardi G.W., Maras M.A.J., Riva Y.R., and Rahman S.F.A. Review of Natural Zeolites and Their Applications: Environmental and Industrial Perspectives. International Journal of Applied Engineering Research 2020 15(7) pp. 730-734.  

Dusselier M. and Davis M.E. Small-Pore Zeolites: Synthesis and Catalysis. Chem. Rev. 2018, 118, 5265−5329. DOI: 10.1021/acs.chemrev.7b00738

  1. There are only few sorption tests to support the affirmation:  “....in addition to favoring the entrapment of PAHs, will allow their separation in water treatment applications” only by calculating the sorption removal

We agree that we should not make unsubstantiated claims and that we can only talk about possible alternatives. This sentence was modified.

Updated version (page 16, lines 512-514)

“…, in addition to favoring the entrapment of PAHs, and its immiscible solid structure pre-sents the adequate characteristics for its separation from aqueous effluents by physical methods.”

  1. It is not shown the sorption capacity obtained with the final synthesized material. Only a figure is presented and this shows only the removal efficiencies. 

We appreciate this observation. The ability of the zeolite to adsorb the functionalized humic acids was included in the results section as well as the capacity of the hybrid sorbent to adsorb the polycyclic aromatic hydrocarbons here studied, each data in the corresponding section.

The adsorption of humic acids in zeolite depends on many factors such as the following: zeolite type, zeolite particle size, pH and concentration of the humic acids in solution. In general, the adsorption of humic acids in zeolite increases with the increasing of the concentration of humic acids in solution (Capasso et al., 2005; Moussavie et al., 2011).

Page 13, lines 424-435. Updated paragraph.

“Studies on the capacity of natural zeolites of different origin to absorb humic acids are extensive. The availability of this type of information allows a better use of these resources in solving environmental pollution problems. In the present work, the mass of HAs sorbed per gram of zeolite was in the range of 4 to 27 mg g-1 using an HA solution of an initial concentration of 30-150 mg L-1. Values of this magnitude have already been reported by Zhan et al. [45] but using a 30 mg L-1 humic acid solution and zeolites (500 mg L-1) treated with HTAB at a concentration of 50 mmol L-1. Capasso et al. [46] reported an HA sorption capacity of 8.5 mg g-1 using Italian natural zeolites at a concentration of 790 ± 76 mg L-1. This value is lower than that obtained in our work, despite the high concentration of HA used. Wang et al. [47] reported a sorption capacity of 37 mg of humic acids in a natural zeolite of Australian origin. These authors used a humic acid solution of 30 mg L-1 and pH = 5. The particular characteristics of this zeolite give it a greater capacity to adsorb humic acids..”

The negative charge of zeolites limits their application to the sorption of cations, however, the treatment of zeolites with surfactants has allowed to extend their application to the removal of nonpolar compounds such as polycyclic aromatic hydrocarbons. The adsorption capacity of non-polar compounds by treated zeolites depends on the type of treatment applied, the type of contaminant, as well as the conditions under which the adsorption process is carried out.

Section "3.6. Sorption of PAH in Funtionalized HA immobilized in zeolite, was reviewed and completed: Pages: 14-15, lines 455-493. In this section a Table was added, Table 3. This Table presents the capacity of the hybrid sorbents to sorb the following PAHs: fluoranthene, anthracene and pyrene. Updated paragraph

“The capacity of the hybrid sorbents evaluated here to adsorb the fluoranthene, an-thracene and pyrene PAHs presented values within the maximum values reported in the literature and even higher. WoÅ‚owiec et al. [48], who tested the capacity of zeolites, or-gano-zeolites and synthetic zeolites (organo-Na-X) to absorb anthracene (initial concen-tration of 20 mg L-1), reported values within the range of 0.08 and 0.14 mg g-1, the highest values of which correspond to the organo-zeolites and organo-synthetic zeolites. The ca-pacity of zeolites and zeolites modified with DDAB, CPC and HDTMA presented an ad-sorbing capacity of 9.7 -10, 9.7-9.9 and 4.7-4.9 µg g-1, to adsorb fluoranthene, anthracene and pyrene, respectively [8].

 The removal of PAHs by natural zeolites has been reported in percentages of 3 - 30% [8]. However, higher removal percentages have been reported when surfactant-modified zeolites were used. Fluoranthene, anthracene and pyrene, for example, were removed in percentages above 93% when using clinoptilolite modified with CPC, DDAB and HDTMA. The initial concentration of PAHs was 0.1, 0.05 and 0.1 mg L-1, respectively, and the ratio between the solid sorbent and the PAH solution was 1:100 [8]. The effect of the particle diameter on the removal of PAH by sorption with organo-zeolite (zeolite treated with stearyldimethylbenzylammoniumchloride (SDBAC)) has been reported by Lemić et al. [11]. These authors found that the organo-zeolite with smaller diameter (0-0.4 mm) fa-vored the total removal of fluoranthene and pyrene added to an aqueous solution at a concentration of 0.01 mg L-1. The PAH removal capacity of this organo-zeolite was of 0.68 and 0.63 mg g-1, respectively. The removal of anthracene (200 pg) with natural zeolite was reported by Manni et al. [49]. When 10 g of zeolite was suspended in 150 mL of the an-thracene solution, 99.7% of PAHs were removed. Plaza et al. [38] reported the removal of pyrene with HA at a concentration of 25 mg L-1. The sorption capacity of HA was 3.4 mg g-1 for an initial concentration of pyrene of 0.11 mg L-1. The removal percentages reported by the cited authors are high; however, it is important to point out the low initial concen-tration of the PAHs, which was below their solubility in water.

Zeolites and HAs are substances that are freely found in nature and have been re-ported to have the ability to trap a wide variety of contaminants [43]. The chemical treat-ment of these substances is a practice commonly used to increase their capacity as sorbents, as well as to extend their application to remove a greater variety of polar and non-polar pollutants [50]. The type of modification will depend on the type of contami-nants to be removed. In the present work, the trapping of hydrocarbons with zeolites re-duced to nanoparticle diameters and the functionalization of humic acids to trap aro-matic-type compounds allowed the elimination of aromatic hydrocarbons with three and four rings. The lower percentage of sorption in the nanoparticles of the fluoranthene has been attributed to their angled molecular structure, which limits their linkage [51]. The crystallinity of water when submerging the solid support HA-TPH-PBr-PAHs allows to assume the irreversibility of the sorption process not only of HA but also of PAHs..”

 Zhan Y., Lin J., Qiu Y., Gao N., Zhu Z.. Adsorption of humic acid from aqueous solution on bilayer hexadecyltrimethyl ammonium bromide-modified zeolite. Front. Environ. Sci. Engin. China 2011, 5(1): 65–75. DOI 10.1007/s11783-010-0277-z

 Capasso, S. Salvestrini, E. Coppola, A. Buondonno, C. Colella, Sorption of humic acid on zeolitic tuff: a preliminary investigation, Appl. Clay Sci. 28 (2005) 159–165.

Wang S., Terdkiatburana T., Tadé M.O., Adsorption of Cu(II), Pb(II) and humic acid on natural zeolite tuff in single and binary systems, Sep. Purif. Technol. 62 (2008) 64–70.

Moussavi G., Talebi S., Farrokhi M., Sabouti R.M.. The investigation of mechanism, kinetic and isotherm of ammonia and humic acid co-adsorption onto natural zeolite. Chemical Engineering Journal 2011. 171: 1159-1169. doi:10.1016/j.cej.2011.05.016

  1. The pH values of 0.5 and 3.5 were applied without having as support a study, a previous study results were used

Tashauoei, H.R.; Movahedian Attar, H.; Amin, M.M.; Kamali, M.; Nikaeen, M.; Vahid Dastjerdi, M., Removal of cadmium and  humic acid from aqueous solutions using surface modified nanozeolite A. Int. J. Environ. Sci. Tech. 2010, 7 (3), 497-508. [https://doi.org/10.1007/BF03326159]

Sorption of humic acids in natural zeolites or treated zeolites has been carried out under different conditions of pH, temperature and humic acids/zeolites ratio, among other factors. However, as far as pH is concerned, it is recommended to carry out the sorption process with humic acids in the solid phase (Kang and Xing 2005; Zeledón-Toruño et al., 2007). For this reason, immobilization was carried out at low pH and using values already reported in the literature. The reference by Tashauoei et al.(2010) was cited as unique, because the authors used Zeolite from Sigma Aldrich although these authors modified the zeolite with HTBA, in addition, the sorption process was carried out at pH=3. The research about the sorption of humic acid in zeolites is really interesting as sorption is dependent on the characteristics and composition of these porous materials.

Regarding polycyclic aromatic hydrocarbons, it has been reported that the pH does not affect their adsorption because they are nonpolar compounds.

In response to the question of reviewer 1, which we appreciate, a paragraph was introduced in the manuscript in order to present some examples of humic acid sorption works in natural zeolites (page 13 lines 424-435). This paragraph was already cited in the response to the question 4.

Some examples of studies about the effect of pH on the sorption of humic acids and PAH are the following:

i. Lin J, Zhan Y (2012) Adsorption of humic acid from aqueous solution onto unmodified and surfactant-modified chitosan/zeolite composites. Chem Eng J 200–202:202–213

“The effect of solution pH on the adsorption of HA on CSZ and SMCSZ was investigated in the pH range 4–12, and the results are shown in Fig. 5. Fig. 5 showed that an increase in solution pH from 4 to 12 caused an obvious decrease in the HA adsorption capacity for CSZ from 63.7 to 5.11 mg/g. This indicates that the adsorption of HA on CSZ is favored at lower pH value.”

ii. Anirudhan T.S., Ramachandran M. Surfactant-modified bentonite as adsorbent for the removal of humic acid from wastewaters. Applied Clay Science 35 (2007) 276–281. doi:10.1016/j.clay.2006.09.009

“The effect of pH on the adsorption of HA onto SMB was studied by varying the pH of the solution from pH 3.0 to 10.0. The results are shown in Fig. 1. It was observed that the maximum percentage removal of HA was at pH 3.0. As the pH of the solution was increased from 3.0 to 10.0, the percentage removal of HA decreased from 99.2 to 15.3% at an initial concentration of 25 μmol/L. The effects of pH on HA adsorption onto SMB are two-fold. The maximum adsorption at pH 3.0 is due the external hydrogen bonds formed between.”

iii. Puszkarewicz A. and Kaleta J. The Efficiency of the Removal of Naphthalene from

Aqueous Solutions by Different Adsorbents. Int. J. Environ. Res. Public Health 2020, 17, 5969; doi:10.3390/ijerph17165969

“The effectiveness of naphthalene adsorption on activated carbons was higher at pH 5 and 7. In an alkaline environment, the reduction of naphthalene occurred to a lesser extent. The weaker adsorption capacity of the activated carbons observed in an alkaline environment at pH = 8, 9 and 10 was the result of the electrostatic repulsion of negative charges located on the surface of the adsorbents and adsorbate. Similar relationships (a decrease of adsorption with increasing pH) were observed on other active carbons [30,31]. In the case of clinoptilolite, the pH did not have a significant effect on the adsorption process.”

iv. Zoraida C. Zeledón-Toruño, Conxita Lao-Luque, F. Xavier C. de las Heras, Montserrat Sole-Sardans. Removal of PAHs from water using an immature coal (leonardite). Chemosphere 67 (2007) 505-512. doi:10.1016/j.chemosphere.2006.09.04

“The sorption of PAHs on leonardite was studied at pH 2, 4, 6 and non-adjusted pH to determine the optimum pH for the sorption of these HOCs. ….. Fig. 1 shows that different initial pH values of the solutions evaluated do not significantly affect the adsorption process of PYR, B(k)F, B(a)P and B(g,h,i)P. Between pH values of 2 and 6, the removal differences were under 9%. This is due to the properties of these chemical compounds, because PAHs are very chemically inert. Their bond linkages give them chemical stability. As a result, heavier PAHs with more C=C bonds are less affected by the pH.”

Kang S. and Xing B. Phenanthrene Sorption to Sequentially Extracted Soil Humic Acids and Humins. Environ. Sci. Technol. 2005, 39, 134-140

  1. No study is presented to compare the sorption capacity and results with other data and studies available in the scientific literature

The response to this question was included in question 4. Table 3 was included. This Table presents the mass of PAHs (fluoranthene, anthracene and pyrene) sorbed per unit mass of hybrid zeolite. These values were compared with data found in the literature. (Pages: 14-15, lines 445-493.)

  1. No tests are available with real wastewater effluents.

We agree with reviewer 1. It is important to conduct the investigation using real contaminated systems. However, it was not considered in our research because of the different types of pollutants that can be found in such systems, which take us away from the objective of our research. In previous works, we applied modified humic acids to remove complex mixtures of recalcitrant hydrocarbons and we worked with a real polluted soil collected near an oil Refinery. (García Díaz et al. 2013)

Hardi et al. (2020), who conducted a review on the application of zeolites in different sectors, concerning the use of zeolites in wastewater treatment problems, wrote the following:

“Removal pollutant from wastewater or gas process needs appropriately modified zeolite. Type of pollutants in wastewater consists of ammonium, phosphorus, heavy metal, inorganic anion, organics, and dye adsorption. Since natural zeolites have some physicochemical properties to adsorb or ion exchange some of compound, natural zeolites have the possibility to remove pollutant from water [9].”

Hardi G.W., Maras M.A.J., Riva Y.R., and Rahman S.F. A Review of Natural Zeolites and Their Applications: Environmental and Industrial Perspectives. International Journal of Applied Engineering Research 2020 15(7):730-734

García-Díaz, C.; Ponce-Noyola, M.T.; Esparza-García, F.; Rivera-Orduña F.; Barrera-Cortés, J. (2013) PAH removal of high molecular weight by characterized bacterial strains from different organic sources. International Journal on Biodeterioration and Biodegradation, 85:311-322. https://doi.org/10.1016/j.ibiod.2013.08.016

  1. No sustainability considerations are made with reference to the materials or process studied, although this is the specific target of this journal

The studies reported on the sorption of different types of pollutants (including hydrocarbons) by means of natural zeolites of different origin are interesting and numerous (Morante-Carballo et al., 2021). The flexibility to change the sorption properties of these materials, either with treatments or by varying the conditions of the sorption process, allows the use of these natural resources in solving current problems of environmental pollution. In addition, using organic fertilizers, such as humic acids, gives the opportunity to degrade the pollutants trapped by these porous materials (García Díaz et al., 2013). The sustainability nature of the work carried out lies precisely in the use of natural materials with selective sorbent properties that allow the remediation of the environment (Chmielevská 2010). See the conclusion.

Morante-Carballo, F.; Montalván-Burbano, N.; Carrión-Mero, P.; Jácome-Francis, K. Worldwide Research Analysis on Natural Zeolites as Environmental Remediation Materials. Sustainability. 2021, 13, 6378 [https://doi.org/10.3390/su13116378]

García-Díaz, C.; Ponce-Noyola, M.T.; Esparza-García, F.; Rivera-Orduña F.; Barrera-Cortés, J. (2013) PAH removal of high molecular weight by characterized bacterial strains from different organic sources. International Journal on Biodeterioration and Biodegradation, 85:311-322

Chmielevská E. Zeolites – Materials of sustainable significance. (Short retrospective and outlook). Environment Protection Engineering 2010 36(4): 127.135

  1. There are missing the authors details, institutions, e-mails, etc. and this is a problem also for assuming the authorship of this study.

The authors appreciate the observation; however, we must mention that this information was included in the manuscript uploaded to the journal in Word format.

Reviewer 2 Report

Dare Editor, thank you for concreting me to review the manuscript number: sustainability-1364027

This paper titled “Polycyclic aromatic hydrocarbons sorption by functionalized humic acids immobilized in micro and nano zeolites"

The manuscript describes  the capacity of functionalized humic acids immobilized in zeolite to absorb some Polycyclic aromatic hydrocarbons.

In my view the manuscript include a critical view of the research area. This manuscript provided some interesting results and useful information. However, there are still some revisions need to be made. Minor points are attached.

Overall, it is a clearly written article and it is concise.

Author Response

The authors appreciate the suggestions and comments made by the reviewers. They allowed us to substantially improve the presentation of the proposed manuscript for its publication in the Sustainability Journal.

The specific questions were answered individually, in order of appearance in the document received with annotations. The corrected manuscript was submitted for review of English grammar.

Reviewer: 2

“ Dear Editor, thank you for concreting me to review the manuscript number: sustainability-1364027

This paper titled “Polycyclic aromatic hydrocarbons sorption by functionalized humic acids immobilized in micro and nano zeolites"

The manuscript describes  the capacity of functionalized humic acids immobilized in zeolite to absorb some Polycyclic aromatic hydrocarbons.

In my view the manuscript include a critical view of the research area. This manuscript provided some interesting results and useful information. However, there are still some revisions need to be made. Minor points are attached.

Overall, it is a clearly written article and it is concise.”

Page 3, line 119: Need reference

Equation 1 needs reference

Equation 2 needs reference

Table 1, Define the abbreviations!!!

Table 1, Complete the statistical analysis

Page 10, Line 357 Needs reference

The authors appreciate the review work done to the manuscript by the reviewer 2, as well as their comments. The manuscript was revised, corrected and sent for revision of the English grammar. The indicated references were included, as well as the omitted definitions of the indicated abbreviations. The statistical analysis of the experimental data presented in Table 1 and Figure 6 was also completed. A supplemental figure (Figure S1) was included in response to the suggested reference on page 9, line 357.

Round 2

Reviewer 1 Report

The manuscript has been improved following the suggestions of the Reviewers